# Regenerative Inflammation: The Mechanism Explained from the Perspective of Buffy-Coat Protagonism and Macrophage Polarization

**DOI:** 10.3390/ijms252011329

**Published:** 2024-10-21

**Authors:** Rubens Andrade Martins, Fábio Ramos Costa, Luyddy Pires, Márcia Santos, Gabriel Silva Santos, João Vitor Lana, Bruno Ramos Costa, Napoliane Santos, Alex Pontes de Macedo, André Kruel, José Fábio Lana

**Affiliations:** 1Medical School, Tiradentes University Center, Maceió 57038-000, Brazil; rubensdeandrade@hotmail.com; 2Department of Orthopedics, FC Sports Traumatology, Salvador 40296-210, Brazil; fabiocosta123@uol.com.br; 3Department of Orthopedics, Brazilian Institute of Regenerative Medicine (BIRM), Indaiatuba 13334-170, Brazil; luyddypires@gmail.com (L.P.); dranapolianesantos@gmail.com (N.S.); alex_macedo@icloud.com (A.P.d.M.); josefabiolana@gmail.com (J.F.L.); 4Regenerative Medicine, Orthoregen International Course, Indaiatuba 13334-170, Brazil; kruel.andre@gmail.com; 5Nutritional Sciences, Metropolitan Union of Education and Culture, Salvador 42700-000, Brazil; marcinha_mairi@hotmail.com; 6Medical School, Max Planck University Center (UniMAX), Indaiatuba 13343-060, Brazil; jvblana@gmail.com; 7Medical School, Zarns College, Salvador 41720-200, Brazil; fabiocosta7113@gmail.com; 8Clinical Research, Anna Vitória Lana Institute (IAVL), Indaiatuba 13334-170, Brazil; 9Medical School, Jaguariúna University Center (UniFAJ), Jaguariúna 13911-094, Brazil

**Keywords:** buffy-coat, macrophage polarization, regenerative inflammation, platelet-rich plasma, mesenchymal stem cells

## Abstract

The buffy-coat, a layer of leukocytes and platelets obtained from peripheral blood centrifugation, plays a crucial role in tissue regeneration and the modulation of inflammatory responses. This article explores the mechanisms of regenerative inflammation, highlighting the critical role of the buffy-coat in influencing macrophage polarization and its therapeutic potential. Macrophage polarization into M1 and M2 subtypes is pivotal in balancing inflammation and tissue repair, with M1 macrophages driving pro-inflammatory responses and M2 macrophages promoting tissue healing and regeneration. The buffy-coat’s rich composition of progenitor cells, cytokines, and growth factors—such as interleukin-10, transforming growth factor-β, and monocyte colony-stimulating factor—supports the transition from M1 to M2 macrophages, enhancing tissue repair and the resolution of inflammation. This dynamic interaction between buffy-coat components and macrophages opens new avenues for therapeutic strategies aimed at improving tissue regeneration and managing inflammatory conditions, particularly in musculoskeletal diseases such as osteoarthritis. Furthermore, the use of buffy-coat-derived therapies in conjunction with other regenerative modalities, such as platelet-rich plasma, holds promise for more effective clinical outcomes.

## 1. Introduction

The buffy-coat is a layer of leukocytes and platelets that forms at the top of a red cell sample after centrifugation of peripheral blood in an anticoagulated container (Figure 1). This blood fraction is rich in progenitor cells—including CD34+ CD133+ erythroid and granulocytic/monocytic progenitors—and plays a crucial role in tissue regeneration and the modulation of the inflammatory response [1]. Buffy-coat cells, particularly monocytes and platelets, have been studied for their ability to modulate the inflammatory response and promote the regeneration of injured tissues [2]. The role of the buffy-coat in tissue regeneration and macrophage polarization stands out for its therapeutic potential in various inflammatory conditions and tissue injuries, paving the way for innovative regenerative therapies that could revolutionize treatment protocols in both acute and chronic conditions [3]. 

Regenerative inflammation is emerging as a critical mechanism in both acute and chronic tissue healing. In orthopedics, conditions such as osteoarthritis, tendinopathies, and muscle injuries are prime targets for regenerative therapies that modulate inflammation [4,5,6]. In cardiovascular medicine, regenerative inflammation plays a pivotal role in heart tissue repair following post-myocardial infarction, where tissue damage can lead to chronic inflammation and fibrosis [7]. Research into the buffy-coat is evolving as its potential in regenerative therapies becomes increasingly apparent. Therefore, understanding its role in tissue regeneration is crucial as medicine moves toward personalized treatment strategies.

A significant phenomenon in the process of immunomodulation is macrophage polarization [3,8], which involves the activation of macrophages into distinct functional subtypes known as M1 and M2 macrophages. M1 macrophages are typically associated with the inflammatory response and the destruction of infectious agents and damaged tissues, while M2 macrophages are responsible for tissue repair and the regulation of the inflammatory response [8]. Regenerative inflammation occurs when there is adequate activation of M2 macrophages, promoting healing and, more importantly, the regeneration of damaged tissues [3].

To better understand the mechanisms involved in regenerative inflammation, and how the buffy-coat and macrophage polarization contribute to this process, several studies have been conducted. These studies have shown that the buffy-coat contains a wide range of soluble factors, such as cytokines and growth factors, that can modulate macrophage polarization and promote tissue regeneration [3,5,9,10]. As shown in Table 1, cytokines play a crucial role in inflammatory response modulation and macrophage activation [11]. Cytokines are signaling molecules secreted by various immune system cells, and among those present in the buffy-coat are transforming growth factor-β, interleukin-10, and monocyte colony-stimulating factor, which have been associated with macrophage polarization to the M2 phenotype and the promotion of tissue regeneration [11].

In addition to cytokines, other factors present in the buffy-coat, such as growth factors, play an essential role in tissue regeneration [11]. Platelet-derived growth factor and vascular endothelial growth factor, for example, have been pointed out as promoters of angiogenesis and vascular regeneration, contributing to the repair of injured tissues [11].

It is crucial to understand the difference between regeneration—the main scope of Regenerative Medicine—and tissue healing. Tissue regeneration refers to the ability of the tissue to reconstruct itself and replace damaged cells with healthy ones, resulting in the restoration of the tissue’s structure and function [12]. On the other hand, tissue healing involves the formation of scar tissue that fills the injured area but cannot fully restore the original structure and function of the tissue [12]. Beyond potential cellular differentiation action, signaling plays a crucial role in regulating macrophage activation and polarization, promoting the transition to the M2 phenotype and, consequently, facilitating tissue regeneration [11,13,14]. Modulating the inflammatory response through signaling from the buffy-coat is fundamental to coordinating the microcellular environment during regeneration, helping to reduce M1 macrophage activity and stimulate M2 macrophage activation and angiogenesis [11]. In short, this signaling is essential for establishing a microenvironment conducive to tissue regeneration. The crosstalk between signaling pathways such as nuclear factor kappa B (NF-Κb) for M1 activation and Janus kinase/signal transducer and activator of transcription (JAK/STAT) and phosphoinositide 3-kinase/Akt/mechanistic target of rapamycin (PI3K/Akt/mTOR) for M2 polarization (Figure 2), illustrates the complexity of immune modulation during tissue regeneration [15]. Buffy-coat-derived cytokines and growth factors may upregulate these pathways and enhance macrophage polarization, thus contributing to accelerated tissue healing [3].

An in-depth understanding of the interaction between the buffy-coat, cytokines, growth factors, and macrophage polarization is essential for developing new therapeutic strategies aimed at effectively modulating the inflammatory response and promoting tissue regeneration in various clinical conditions. Therefore, continuing studies in this area are fundamental to identifying therapeutic targets and developing innovative approaches in the field of regenerative inflammation.

## 2. Cellular Composition of the Buffy-Coat and Its Mononuclear Cells

The buffy-coat comprises various cells, including lymphocytes, monocytes, eosinophils, basophils, and hematopoietic progenitor cells [16,17]. However, mononuclear cells are particularly noteworthy due to their crucial role in modulating the inflammatory response and promoting tissue regeneration. Mononuclear cells are primarily composed of lymphocytes and monocytes, the latter being of particular interest due to their ability to polarize into M2 macrophages, which are fundamental to promoting tissue regeneration [9,18].

Monocytes, upon recruitment to injury sites, can differentiate into macrophages, contributing to both the inflammatory and tissue repair phases. Their differentiation is influenced by the buffy-coat, which contains factors like transforming growth factor-β (TGF-β) and interleukin-10 (IL-10) [11]. This polarization is essential for promoting an anti-inflammatory environment and regenerating injured tissues [11]. Additionally, the presence of hematopoietic progenitor cells in the buffy-coat is significant, as these immature cells can develop into all types of blood cells, including leukocytes, erythrocytes, and platelets, contributing to tissue regeneration and repair [11,17,19]. Therefore, the cellular composition of the buffy-coat, with an emphasis on mononuclear cells, plays a fundamental role in modulating the inflammatory response and promoting tissue regeneration [9].

A thorough analysis of the cellular and molecular composition of the buffy-coat and the role of mononuclear cells in inflammatory and regenerative processes may offer valuable insights for developing targeted therapeutic strategies aimed at optimizing the inflammatory response and stimulating tissue regeneration in a clinical context. To further clarify the cellular composition of the buffy-coat and its respective contributions to tissue regeneration and inflammation, a summary of the key cellular components is provided in Table 2. This table outlines the specific roles of each cell type in the regenerative process.

## 3. Macrophage Polarization in Regenerative Inflammation

Regenerative inflammation is a complex process involving the interaction of different cell types and inflammatory mediators, resulting in the regeneration and repair of injured tissues [20]. At its core, regenerative inflammation is an orchestrated effort to balance the pro-inflammatory signals necessary for pathogen and damaged tissue clearance with the reparative signals that drive tissue reconstruction [21]. This balance ensures that inflammation does not escalate into chronic states, which can be detrimental to healthy tissue [21]. The early inflammatory response, dominated by M1 macrophages, is followed by a carefully timed switch to M2 macrophages to promote repair and regeneration [21].

M1 macrophages, also called classically activated macrophages, mainly exert pro-inflammatory effects. M1 macrophages are activated by toll-like receptors, which are triggered by pathogen-associated molecular patterns (PAMPs) such as lipopolysaccharide (LPS). Cytokines like interferon-γ and TNF-α, meanwhile, activate macrophages through their specific cytokine receptors, further contributing to the inflammatory response [13]. Regarding the pro-inflammatory mechanism, M1 macrophages secrete a series of cytokines, including IL-1, IL-6, IL-12, IL-23, and TNF-α, released by TH1 cells [22]. Subsequently, M1 macrophages trigger various immune responses, including pathogen destruction and the initiation of inflammation. Finally, a large-scale inflammatory response arises [13]. It is worth noting that an excessive inflammatory response caused by M1 macrophages usually damages healthy tissues, as seen in conditions like rheumatoid arthritis, where it manifests in patients as joint pain [23]. However, it is essential to emphasize that although they are primarily recognized for their pro-inflammatory effects, M1 macrophages also play a role in driving the adaptive immune response [23]. 

The NF-κB signaling pathway plays a central role in the activation of M1 macrophages by promoting the expression of pro-inflammatory cytokines such as IL-1, TNF-α, and IL-6 [24]. Upon activation, toll-like receptors (TLRs) on macrophages detect PAMPs, leading to NF-κB activation, which drives the early inflammatory response necessary for pathogen clearance and damaged tissue removal [24]. Once the NF-κB pathway is activated, a cascade of transcription factors is initiated, which enhances the expression of genes involved in inflammation [25]. Notably, chronic activation of NF-κB, as observed in conditions like rheumatoid arthritis and chronic tendinopathies, can lead to a persistent pro-inflammatory state that impedes tissue repair and regeneration [24]. 

While this pathway is crucial during the initial phases of inflammation, problems arise when it becomes dysregulated, leading to chronic inflammation and tissue damage [24]. 

Polarizing macrophages to the M2 phenotype is critical for resolving inflammation and promoting tissue regeneration [23]. This process is driven by cytokines present in the buffy-coat, such as TGF-β and IL-10, which specifically modulate macrophage behavior toward a reparative phenotype [23]. These factors are essential for promoting an anti-inflammatory environment, stimulating angiogenesis, and contributing to the repair of injured tissues. The polarization of macrophages to the M2 phenotype is mediated by the binding of specific receptors, such as the IL-13 and IL-4 receptors, which promote the production of anti-inflammatory factors such as IL-10 and TGF-β [8,13]. The polarization of macrophages to the M2 phenotype is also associated with producing factors that promote wound healing, such as growth factors, extracellular proteins, and enzymes involved in tissue remodeling [8]. Additionally, M2 macrophage polarization also plays an essential role in the adaptive immune response by regulating T regulatory lymphocyte activation and secreting cytokines that contribute to the immune response [26]. 

The switch from M1 to M2 macrophages is a pivotal moment in tissue repair. If the M1-to-M2 transition is delayed or incomplete, chronic inflammation can ensue, leading to tissue scarring, fibrosis, or even permanent tissue damage [27]. On the other hand, enhanced M2 polarization not only reduces inflammation but also plays a crucial role in angiogenesis, the formation of new blood vessels, which is essential for delivering nutrients and oxygen to the regenerating tissue [27]. Additionally, M2 macrophages release matrix metalloproteinases (MMPs), which remodel the extracellular matrix, a necessary step in tissue regeneration [27]. 

The JAK/STAT pathway is pivotal in mediating M2 polarization. Upon the binding of IL-4 and IL-13 to their receptors, JAK proteins are activated, which in turn phosphorylate STAT proteins [15]. STAT6 is particularly important for M2 polarization, as it induces the expression of genes involved in anti-inflammatory responses and tissue repair, including those coding for IL-10 and TGF-β [28]. This pathway not only promotes tissue regeneration but also contributes to the resolution of inflammation by suppressing M1-associated pro-inflammatory responses [29,30]. Recent findings also suggest that the JAK/STAT pathway plays a role in maintaining the macrophage phenotype during prolonged tissue repair, particularly in chronic conditions [31]. By inhibiting STAT1 (which supports M1 polarization) and favoring STAT6 activation (which drives M2 polarization), the JAK/STAT axis effectively shifts the macrophage response toward a more regenerative state [31]. This highlights the therapeutic potential of modulating the JAK/STAT pathway, especially in chronic inflammatory diseases like osteoarthritis or fibrosis.

The signaling pathways involved in macrophage polarization play a fundamental role in regulating the immune response and promoting tissue regeneration. These pathways include NF-κB (nuclear factor kappa B), MAPK (mitogen-activated protein kinases), TGF-β (transforming growth factor beta), JAK/STAT (Janus kinase/signal transducer and activator of transcription), PI3K/Akt/mTOR (phosphoinositide 3-kinase/Akt/mechanistic target of rapamycin), and NLRP3 [13].

Among these, the PI3K/Akt/mTOR pathway is significant for M2 polarization and tissue repair. Activation of this pathway promotes cell survival and proliferation, essential for the resolution of inflammation and tissue regeneration [32]. By regulating the transition from M1 to M2 macrophages, the PI3K/Akt/mTOR pathway facilitates the establishment of a regenerative microenvironment [32]. This balance between inflammation and repair is critical in clinical applications, where excessive inflammation can hinder recovery, while effective M2 activation promotes healing [32,33].

Furthermore, the PI3K/Akt/mTOR pathway is involved in metabolic reprogramming of macrophages. M2 macrophages rely on oxidative phosphorylation, a more efficient energy production pathway, in contrast to M1 macrophages, which depend on glycolysis [34]. This metabolic shift ensures that M2 macrophages have the energy required to sustain tissue repair activities over an extended period. By modulating cellular metabolism, the PI3K/Akt/mTOR pathway not only drives M2 polarization but also ensures that these macrophages can perform their regenerative functions optimally [34]. 

The activation of these pathways is involved in the response of macrophages to stimuli in their microenvironment, influencing the phenotype shift of macrophages to pro-inflammatory (M1) or anti-inflammatory (M2) phenotypes [13]. For example, the NF-κB pathway plays a crucial role in activating pro-inflammatory genes in M1 macrophages, while the PI3K/Akt/mTOR pathway has been associated with promoting polarization to the M2 phenotype, contributing to tissue regeneration and resolving the inflammatory response [24,32]. Therefore, identifying therapeutic targets in these signaling pathways and developing innovative strategies for their modulation represents a promising area of research with significant clinical applications. Emerging therapies, including those involving platelet-rich plasma (PRP), stem cells, and pharmacological inhibitors, are being explored to modulate these pathways, offering new hope for enhancing tissue regeneration in chronic inflammatory conditions. Mediating the adequate polarization of M1 and M2 macrophages is crucial to achieving the necessary balance between the inflammatory response and tissue regeneration [14].

## 4. Difference Between Leukocyte-Rich and Leukocyte-Poor Platelet-Rich Plasma

PRP can be classified into two main types: leukocyte-rich PRP (L-PRP) and leukocyte-poor PRP (P-PRP). L-PRP contains a significant amount of leukocytes, while P-PRP has a low concentration of these cells. This difference in cellular composition directly affects inflammatory responses and tissue regeneration processes [5,9]. Leukocytes can secrete cytokines and other inflammatory mediators that play an essential role in the inflammatory response [6,13]. Additionally, leukocytes can interact with platelets and modulate the release of growth factors by platelets [35]. These differences in PRP composition can influence macrophage polarization and their regenerative capacity in the inflammatory environment [5,9]. More information regarding these two preparations can be found in Table 3.

The use of PRP with different levels of leukocytes is a subject of ongoing debate in the field of healing and regenerative medicine, as researchers and clinicians continue to explore the optimal balance of components that maximize therapeutic outcomes. It is argued that leukocytes present in PRP can have a positive impact on regenerative inflammation by releasing pro- and anti-inflammatory molecules [9]. However, there are concerns about the potential of leukocytes, particularly neutrophils, to exacerbate the inflammatory response through the release of harmful cytokines and metalloproteinases [33]. Moreover, it is questioned whether the potential catabolic effects of L-PRP outweigh its antimicrobial and reparative signaling benefits. The debate over whether to enrich PRP with leukocytes continues to pose a challenge in standardizing this orthobiologic product. 

L-PRP is characterized by its higher concentration of leukocytes, including neutrophils and monocytes, which can produce both pro-inflammatory and anti-inflammatory cytokines [5,9,36,37]. This composition makes L-PRP suitable for situations where an initial inflammatory response is necessary, such as in acute tendon injuries or infections [37,38,39]. Neutrophils play a key role in the early stages of healing by releasing MMPs, which degrade damaged tissue, thereby promoting tissue remodeling and regeneration [40]. However, the excessive release of pro-inflammatory cytokines, such as TNF-α and IL-1, by L-PRP may sometimes lead to an exaggerated inflammatory response, potentially delaying the healing process [41]. Despite some controversy, platelet derivatives rich in leukocytes can be (and have been) utilized in post-surgical applications, such as rotator cuff repair, where the inflammatory properties help facilitate initial tissue attachment and prevent infections [42,43]. However, it is important to remember that the inflammatory response must be carefully managed in order to prevent prolonged inflammation that could lead to scarring or delayed recovery.

P-PRP, on the other hand, contains fewer leukocytes and is preferred in scenarios where the goal is to minimize inflammation, such as in chronic tendinopathies or osteoarthritis [44,45]. The lower concentration of leukocytes in P-PRP reduces the secretion of pro-inflammatory mediators, thereby lowering the risk of excessive inflammation [46]. Instead, P-PRP is rich in growth factors like vascular endothelial growth factor (VEGF) and platelet-derived growth factor (PDGF), which promote angiogenesis, collagen synthesis, and tissue regeneration without inducing significant inflammation [46,47]. This makes P-PRP an ideal option for chronic inflammatory conditions, where sustained inflammation can lead to further tissue damage.

In chronic conditions such as osteoarthritis, where inflammation plays a key role in the ongoing degeneration of cartilage, P-PRP has been shown to offer significant therapeutic benefits [48,49,50]. A study demonstrated that P-PRP injections are superior in terms of clinical improvement in comparison to hyaluronic acid injections or oral non-steroidal anti-inflammatory drug administration in knee OA patients followed up for 52 weeks [48]. By reducing inflammation and promoting cartilage regeneration, P-PRP may delay the need for invasive procedures like joint replacement surgeries [50,51]. P-PRP is also commonly used for chronic tendinopathies, where inflammation persists over extended periods and hinders the natural healing process [52].

Given these differences, selecting the appropriate PRP formulation depends on the clinical scenario. In acute injuries, L-PRP may be advantageous due to its ability to rapidly mobilize the immune response and clear damaged tissues. Conversely, in chronic conditions where inflammation needs to be controlled, P-PRP may be the preferable option for its ability to promote healing with minimal inflammatory activity.

Considering the previously discussed information, a potential approach to optimizing PRP therapy could involve administering L-PRP during the acute inflammatory phase and P-PRP during the tissue remodeling phase. This strategy could theoretically leverage the strong inflammatory response induced by L-PRP for early tissue clearance and healing, while P-PRP may reduce prolonged inflammation and support tissue regeneration in the later stages. Such a dual-phase treatment could be beneficial to managing both acute injuries and underlying chronic conditions, such as tendon ruptures in patients with tendinopathy. While this concept is not yet fully explored, it aligns with the distinct roles that different PRP formulations play in modulating inflammation and repair.

Despite the lack of consensus, the scientific literature suggests that the presence of leukocytes in PRP formulations may play significant roles in the inflammatory response and tissue regeneration processes [5,6,9,53,54,55]. In clinical settings, L-PRP is often employed in acute injuries, such as tendon or muscle tears, where a stronger initial inflammatory response is desired to promote early healing. Additionally, L-PRP proves beneficial in surgical contexts, such as rotator cuff repairs, where the pro-inflammatory properties of leukocytes help facilitate tissue attachment and prevent infections. However, managing inflammation carefully is crucial to avoid excessive tissue damage. Conversely, P-PRP is better suited for chronic conditions, such as osteoarthritis and tendinopathy, where prolonged inflammation could impede healing. The lower leukocyte concentration in P-PRP promotes tissue regeneration while minimizing inflammation, offering therapeutic benefits in long-term conditions. Clinicians must tailor the use of L-PRP or P-PRP based on patient-specific needs, balancing inflammation with tissue repair to optimize healing outcomes.

Continued research into the precise mechanisms of action for each PRP type will be essential to improve their efficacy in regenerative therapies. Future advancements may involve personalized PRP formulations tailored to the patient’s specific inflammatory status and healing requirements. By tailoring PRP to the individual, clinicians could optimize therapeutic outcomes while minimizing potential side effects, such as excessive inflammation or delayed recovery.

## 5. Peripheral Blood-Derived Mesenchymal Stem Cells

Mesenchymal stem cells (MSCs) have been recognized for their potential in regenerative medicine, including treatments for osteoarthritis. MSCs play a key role in modulating the immune system and possess anti-inflammatory properties that are beneficial in mitigating numerous health conditions, especially musculoskeletal diseases like OA [22,56]. The ability of MSCs to directly manipulate macrophage polarization further contributes to reducing inflammation and enhancing tissue repair within the joint [56,57,58,59].

MSCs derived from peripheral blood, as opposed to bone marrow or adipose-derived MSCs, offer a less invasive collection method and may circulate systemically to sites of injury or inflammation [60]. Once mobilized to these regions, MSCs exert their therapeutic effects through various signaling mechanisms, including the secretion of paracrine factors [61]. Chemokine signaling, especially through interactions between CXCR4 and SDF-1, helps guide MSCs to areas of tissue damage, where they contribute to the resolution of inflammation and tissue repair [62]. 

In the context of degenerative musculoskeletal conditions such as OA, the immunomodulatory effect of MSCs is crucial. Macrophages play a significant role in osteoarthritic inflammation and cartilage degradation, which means that regulating their activity can be a key therapeutic strategy to mitigate symptoms and disease progression [63]. MSCs are known to secrete immunomodulatory factors like interleukin-10, transforming growth factor-β, and prostaglandin E2, all of which help shift macrophage polarization from the pro-inflammatory M1 phenotype to the anti-inflammatory M2 phenotype [64]. This polarization is a necessary step in the reduction of the inflammatory burden within the joint and in the promotion of a healing environment that supports cartilage regeneration and decreases pain.

It is essential to highlight this signaling power of MSCs, since they are classically referenced and mainly renowned for their differentiation potential [56]. However, much of their therapeutic benefit now appears to come from their paracrine activity, whereby MSCs release a variety of bioactive molecules that modulate the local microenvironment [61]. These factors include growth factors such as vascular endothelial growth factor (VEGF), insulin-like growth factor-1 (IGF-1), and hepatocyte growth factor (HGF), all of which play important roles in promoting angiogenesis, reducing inflammation, and facilitating tissue repair [61]. Recently, the ability of MSCs to act as “conductors” within the injured tissue has been emphasized. These cells orchestrate the behavior of other immune and structural cells to optimize the regenerative process [65,66]. 

Another emerging area of research involves the interaction of MSCs with other immune cells, particularly T regulatory (Treg) cells. MSCs promote the expansion of Tregs, which are known to suppress immune responses and maintain immune tolerance, further reducing inflammation in tissues affected by degenerative conditions like OA [67]. By enhancing Treg activity, MSCs may create a more stable, anti-inflammatory microenvironment that supports long-term tissue repair. This immunomodulatory property would be particularly valuable in chronic inflammatory conditions, for example, where immune dysregulation is a contributing factor to disease progression.

The clinical applications of MSCs are vast and continue to grow. In OA treatment, MSCs have been shown to improve cartilage quality and reduce pain levels in patients [68,69]. Clinical trials have demonstrated that intra-articular injections of MSCs lead to significant improvements in joint function [70,71,72,73,74,75]. This could potentially delay the need for joint replacement surgery. MSCs are also being investigated for their use in other musculoskeletal injuries, such as tendon and ligament damage, where their ability to modulate the immune response and promote tissue regeneration is being leveraged to enhance healing outcomes [76,77,78]. 

Overall, the use of MSCs, particularly in conjunction with other regenerative therapies, such as PRP, is a rapidly advancing area of research. The combination of MSCs and PRP could provide synergistic benefits, as PRP can provide growth factors that enhance MSC viability and activity, while MSCs offer the immunomodulatory and regenerative capabilities needed to optimize healing. This combinatorial approach should be further investigated in clinical trials to treat complex musculoskeletal injuries and chronic inflammatory diseases.

## 6. Conclusion

Numerous PRP formulations play fundamental roles in tissue regeneration and the modulation of the inflammatory response. The presence of leukocytes in PRP can directly influence the inflammatory cascade, with effects varying depending on the leukocyte concentration and the specific clinical application. While leukocytes can promote early inflammation crucial for clearing damaged tissue, their excessive presence can potentially prolong inflammation and delay healing. This underscores the urgent need for standardization in PRP therapies, particularly in determining the ideal concentrations of leukocytes and platelets for specific clinical scenarios. Future research should focus on refining these formulations to optimize their efficacy in both acute and chronic conditions.

On the other hand, peripheral blood-derived MSC populations display potent anti-inflammatory and immunomodulatory properties, making them key players in modulating macrophage polarization. By promoting the transition from pro-inflammatory M1 macrophages to anti-inflammatory M2 macrophages, MSCs can effectively contribute to tissue repair and the resolution of inflammation. Their role is especially important in treating chronic inflammatory conditions, such as osteoarthritis, where prolonged inflammation leads to tissue degradation.

An in-depth understanding of the interactions between PRP formulations, MSCs, and macrophage polarization is essential for developing targeted and effective therapeutic approaches. These strategies could significantly enhance tissue regeneration while balancing the inflammatory response, offering new avenues for treatment across a wide range of clinical contexts, from musculoskeletal injuries to chronic degenerative diseases. 

Future therapeutic advancements will likely involve integrating these regenerative strategies with other emerging technologies, such as biomaterials and gene therapy, to further enhance the precision and efficacy of treatments. By combining various approaches, clinicians can better address the complex interplay between inflammation and tissue repair, ultimately improving long-term clinical outcomes.

## Figures and Tables

**Figure 1 ijms-25-11329-f001:**
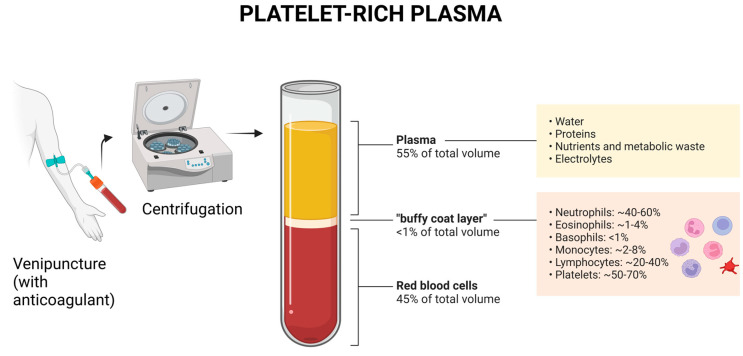
Overview of Platelet-Rich Plasma Preparation. Blood is collected and centrifuged to separate into three layers: red blood cells at the bottom, platelet-rich plasma at the top, and the buffy coat in the middle. This figure shows the approximate cellular composition of the buffy coat, highlighting the distribution of neutrophils, eosinophils, basophils, monocytes, lymphocytes, and platelets.

**Figure 2 ijms-25-11329-f002:**
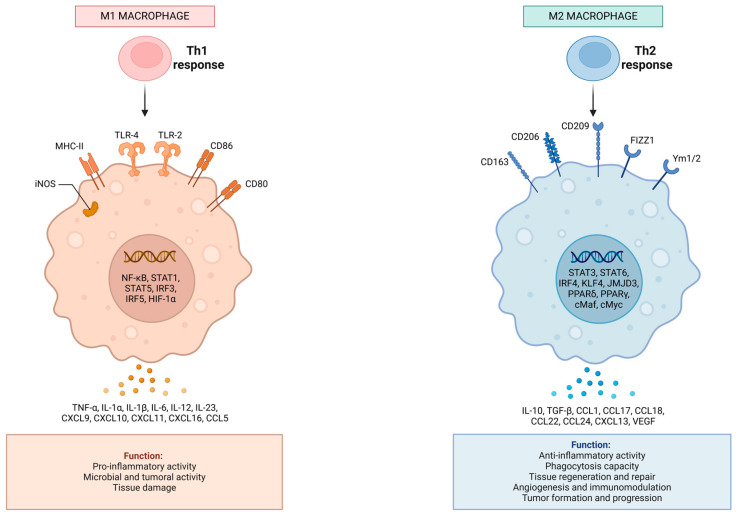
Macrophage Polarization Pathways. Illustration of signaling pathways involved in macrophage polarization. The NF-κB pathway predominantly supports M1 macrophage activation, promoting pro-inflammatory responses. In contrast, the PI3K/Akt/mTOR and JAK/STAT pathways facilitate M2 polarization, contributing to anti-inflammatory effects and tissue repair. The figure highlights the dynamic interactions between these pathways in response to buffy-coat-derived cytokines and growth factors, which play a pivotal role in the balance between inflammation and regeneration.

**Table 1 ijms-25-11329-t001:** Key cytokines and growth factors involved in macrophage polarization.

Factor	Source	Effect on Macrophages	Role in Inflammation/Regeneration
Interleukin-10 (IL-10)	Buffy-Coat, MSCs	Promotes M2 polarization	Anti-inflammatory, tissue regeneration
Transforming Growth Factor-β (TGF-β)	Buffy-Coat, MSCs	Promotes M2 polarization	Regeneration, anti-inflammatory, wound healing
Monocyte Colony-Stimulating Factor (M-CSF)	Buffy-Coat, MSCs	Promotes M2 polarization	Anti-inflammatory, promotes tissue repair
Interferon-γ (IFN-γ)	Th1 cells, M1 macrophages	Promotes M1 polarization	Pro-inflammatory, pathogen clearance
Tumor Necrosis Factor-α (TNF-α)	M1 macrophages	Enhances M1 polarization	Pro-inflammatory, tissue destruction
Platelet-Derived Growth Factor (PDGF)	Platelets	Angiogenesis, supports tissue repair	Tissue regeneration, vascular regeneration
Vascular Endothelial Growth Factor (VEGF)	Platelets, MSCs	Angiogenesis	Promotes vascularization and wound healing

**Table 2 ijms-25-11329-t002:** Cell types composing the buffy-coat and their functions.

Cell Type	Description	Function in Regenerative Inflammation
Leukocytes	White blood cells, including neutrophils, lymphocytes, monocytes	Crucial in the immune response, modulating inflammation and promoting tissue repair
Monocytes	A type of leukocyte that differentiates into macrophages	Involved in macrophage polarization (M1 pro-inflammatory, M2 anti-inflammatory) and tissue regeneration
Lymphocytes	Includes T cells and B cells	Play a role in adaptive immune responses and help regulate inflammation
Platelets	Blood clotting cells	Release growth factors (e.g., PDGF, VEGF) that promote angiogenesis and tissue healing
Progenitor Cells	Precursor cells, including CD34+ progenitors	Contribute to tissue regeneration by differentiating into various cell types and aiding in repair

**Table 3 ijms-25-11329-t003:** Comparison of Leukocyte-Rich PRP (L-PRP) and Leukocyte-Poor PRP (P-PRP) in Regenerative Therapy.

Characteristic	Leukocyte-Rich PRP (L-PRP)	Leukocyte-Poor PRP (P-PRP)
Leukocyte Concentration	High	Low
Inflammatory Response	Stronger, due to leukocytes	Milder
Cytokine Production	Pro-inflammatory (TNF-α, IL-1)	Anti-inflammatory (IL-10)
Macrophage Polarization	Favors M1 polarization	Favors M2 polarization
Tissue Healing Potential	Moderate, due to inflammatory activity	High, promotes tissue regeneration
Best Applications	Acute injuries, infections	Chronic inflammation, soft tissue repair
Potential Side Effects	Excessive inflammation, tissue damage	Minimal, with focused regenerative effect

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
