# Peer review of "Regenerative Inflammation: The Mechanism Explained from the Perspective of Buffy-Coat Protagonism and Macrophage Polarization"

_ijms, 2024, doi:10.3390/ijms252011329_

Round 1

Reviewer 1 Report

Comments and Suggestions for Authors

In this manuscript, Martins RA and collaborators analysed the critical role of the buffy-coat in influencing macrophage polarization and its therapeutic potential. 

Specific comments:

1)       Figures legend should be added.

2)       Section 2: a new table summarizing the cellular composition of the Buffy-Coat and its mononuclear cells should be included.

3)       Line 127-129 sentence is repeated at lines 176-178.

4)       Acronyms need to be checked.

5)       Section 4: applications of L-PRP and P-PRP and their importance on clinical uses should be discussed.

6)       Line 315-317: references are missing.

Comments on the Quality of English Language

 Minor editing of English language revision is required.

Reviewer 2 Report

Comments and Suggestions for Authors

In this review manuscript, Martins et al. give a detailed overview not only over the buffy coat itself, but also over its various cellular and non-cellular components. They mainly focus on the therapeutic opportunities, which lie in these components but also give detailed explanation about the single cell fractions and their responses in this context. In general, this review is exhaustive and highly informative, even for readers, which are not familiar with this research field. The figures are simple, but self-explanatory and support the text.

Overall the writing, phrasing and grammar of the manuscript are excellent and understandable. Very well done. All of the topics are very detailed and sufficiently explained. I have only two minor suggestions, however they should not hinder the publication of this remarkable review manuscript.

Minor points:

Figure 1 is very nice and understandable. It could be slightly improved by adding the approximate percentual amounts of the single contents. This would be especially interesting for the immune cells in the buffy coat.

In lines 151-152: A little error sneaked in in these lines, which state “M1 macrophages are activated by toll-like receptors stimulated by factors such as interferon-γ, LPS (lipopolysaccharide), and TNF-α [13]. This is not correct. Toll-like receptors are activated by LPS and other PAMPs, but not by interferons or TNF, which have their very won receptors. The authors wrote it correctly a few lines later (164-165).
